# TORCHGEO: DEEP LEARNING WITH GEOSPATIAL DATA

## ABSTRACT

Remotely sensed geospatial data are critical for applications including precision agriculture, urban planning, disaster monitoring and response, and climate change research, among others. Deep learning methods are particularly promising for modeling many remote sensing tasks given the success of deep neural networks in similar computer vision tasks and the sheer volume of remotely sensed imagery available. However, the variance in data collection methods and handling of geospatial metadata make the application of deep learning methodology to remotely sensed data nontrivial. For example, satellite imagery often includes additional spectral bands beyond red, green, and blue and must be joined to other geospatial data sources that can have differing coordinate systems, bounds, and resolutions. To help realize the potential of deep learning for remote sensing applications, we introduce TorchGeo, a Python library for integrating geospatial data into the PyTorch deep learning ecosystem. TorchGeo provides data loaders for a variety of benchmark datasets, composable datasets for generic geospatial data sources, samplers for geospatial data, and transforms that work with multispectral imagery. TorchGeo is also the first library to provide pre-trained models for multispectral satellite imagery (e.g. models that use all bands from the Sentinel 2 satellites), allowing for advances in transfer learning on downstream remote sensing tasks with limited labeled data. We use TorchGeo to create reproducible benchmark results on existing datasets and benchmark our proposed method for preprocessing geospatial imagery on-the-fly. TorchGeo is open source and available on GitHub: `redacted`.

## 1 INTRODUCTION

With the explosion in availability of satellite and aerial imagery over the past decades, there has been increasing interest in the use of imagery in remote sensing (RS) applications. These applications range from precision agriculture [39] and forestry [34], to natural and man-made disaster monitoring [60], to weather and climate change [48]. At the same time, advancements in machine learning (ML), larger curated benchmark datasets, and increased compute power, have led to great successes in domains like computer vision (CV), natural language processing (NLP), and audio processing. However, the wide-spread success and popularity of machine learning—particularly of deep learning methods—in these domains has not fully transferred to the RS domain, despite the existence of petabytes of freely available satellite imagery and a variety of benchmark datasets for different RS tasks. This is not to say that there are not successful applications of ML in RS, but that the full potential of the intersection of these fields has not been reached. Indeed, a recent book by Camps-Valls et al. [8] thoroughly details work at the intersection of deep learning, geospatial data, and the Earth sciences. Increasing amounts of research on self-supervised and unsupervised learning methods specific to remotely sensed geospatial imagery [35; 3; 29] bring the promise of developing generic models that can be tuned to various downstream RS tasks. Recent large-scale efforts, such as the creation of a global 10 m resolution land cover map [31] or the creation of global 30 m forest maps [47], pair the huge amount of available remotely sensed imagery with modern GPU accelerated models. To reach the full joint potential of these fields, we believe that we need tools for facilitating research and managing the complexities of both geospatial data and modern machine learning pipelines. We describe the challenges of this below, and detail our proposed solution, TorchGeo.

One major challenge in many RS tasks is the large amount of diversity in the content of geospatial imagery datasets compared to datasets collected for traditional vision applications. For example, most conventional cameras capture 3-channel RGB imagery, however most satellite platforms capture different sets of spectral bands. The Landsat 8 satellite [50] collects 11 bands, the Sentinel-2 satellites [16] collect 12 bands, and the Hyperion satellite [44] collects 242 (hyperspectral) bands all measuring different regions

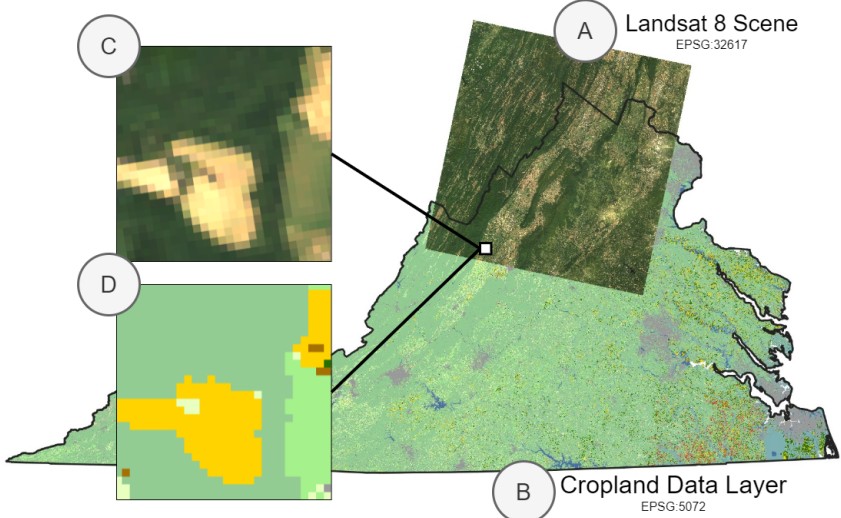

Figure 1: An illustration of the challenges in sampling from heterogeneous geospatial data layers. (**A** and **B**) show example geospatial data layers that a user may want to sample pixel-aligned data from. As these layers have differing coordinate reference systems, patches of imagery (**C** and **D**) sampled from these layers that cover the same area will not be pixel-aligned. TorchGeo transparently performs the appropriate alignment steps (reprojecting and resampling) during data loading such that users can train deep learning models without having to manually align the data layers.

of the electromagnetic spectrum. The exact wavelengths of the electromagnetic spectrum captured by each band can range from 400 nm to 15 $\mu$m. In addition, different sensors capture imagery at different spatial resolutions: satellite imagery can range from 4 km/px (GOES [43]) to 30 cm/px (Maxar WorldView satellites [56]), while imagery captured from drones can go as low as 7 mm/px [1]. Depending on the type of orbit a satellite is in, imagery can be continuous (for geostationary orbits) or daily to biweekly (for polar, sun-synchronous orbits). Machine learning models or algorithms developed for one of these platforms will not generalize across inputs collected by the others, and, as a consequence of this, it is not possible to publish a single set of pre-trained model weights that span imaging platforms. In contrast, ImageNet [15] pretrained models have been proven to be useful in a large number of transfer learning tasks [65]. Researchers and practitioners can often start with ImageNet pre-trained models in a transfer learning setup when presented with vision problems to reduce the overall amount of effort needed to solve the problem. Further, it isn't clear whether the inductive biases built into common modeling approaches for vision problems are immediately applicable to remotely sensed imagery. Large neural architecture search efforts [32; 72] produce models that are optimized for, and indeed, outperform hand designed architectures on vision tasks, but it is an open question whether these transfer to remotely sensed imagery.

Most machine learning libraries have not been designed to work with geospatial data. For example, the Python Imaging Library (PIL) [11], used by many libraries to load images and perform data augmentation, does not support multispectral imagery. Similarly, deep learning models implemented by the torchvision library only support 3 channel (RGB) inputs, and must be adapted or re-implemented to support multispectral data. Datasets of geospatial data can be made up of a heterogenous mix of files with differing file formats, spatial resolutions, projections, and coordinate reference systems (CRS). Libraries such as rasterio [22] and fiona [21] can interface with most types of geospatial data, however further abstractions for using such data in arbitrary deep learning pipelines are limited. Indeed, the gap between loading geospatial data from disk, and using it in a modeling pipeline, is large for all of the reasons mentioned above. As illustrated in Figure 1, users will often need pixel-aligned crops from multiple layers of data: imagery from different points in time over the same space, imagery and corresponding label masks, high-resolution and low resolution imagery from the same space, etc. In contrast, there are a wide variety of software libraries at the intersection of machine learning and other domains. The PyTorch ecosystem alone has torchvision [38], torchtext [67], torchaudio [63], Hugging Face's Transformers [62], PyTorch Geometric [18], TorchMeta [14], and PyTorch Video [17], that provide the tools necessary for abstracting the peculiarities of domain-specific data away from the details of deep learning training pipelines.

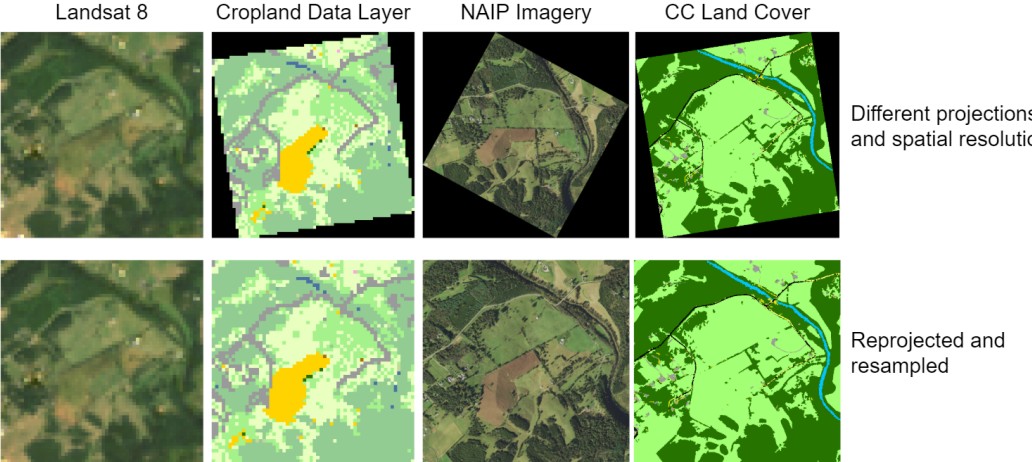

| Landsat 8 | Cropland Data Layer | NAIP Imagery | CC Land Cover |
| --- | --- | --- | --- |

Different projections and spatial resolution

Reprojected and resampled

Figure 2: Different layers of geospatial data often have differing coordinate reference systems and spatial resolutions. (**Top row**) The same physical area cropped from four raster layers with different coordinate reference systems and spatial resolutions—this data is not pixel-aligned and cannot yet be used in modelling pipelines. (**Bottom row**) The same data as after reprojecting into the same coordinate system and resampling to the highest spatial resolution—this data is pixel aligned and can serve as inputs or masks to deep neural networks.

To address these challenges we propose TorchGeo, a Python package that allows users to transparently use heterogenous geospatial data in PyTorch-based deep learning pipelines. Specifically, TorchGeo provides:

1. data loaders for common geospatial datasets from the literature
2. data loaders for combining arbitrary geospatial raster and vector data layers with the ability to sample pixel-aligned patches on-the-fly
3. augmentations that are appropriate for multispectral imagery and geospatial data
4. data samplers appropriate for geospatial data
5. pre-trained models for many common remotely sensed imagery sources

We formally describe TorchGeo, propose and test methods for sampling from large geospatial datasets, and test the effect of ImageNet pretraining versus random weight initialization on several benchmark datasets. We achieve close to state-of-the-art results on all experimental datasets, despite focusing only on creating simple and reproducible results to serve as baselines for future work to build on. We further find that ImageNet pre-training significantly improves spatial generalization performance in a land cover mapping task. We believe that these results are interesting in their own right, and that they highlight the importance of TorchGeo to the larger machine learning community. TorchGeo is open source and available on GitHub: `redacted`.

## 2 DESIGN

Remotely sensed imagery datasets are usually formatted as *scenes*, i.e. tensors $X \in \mathbb{R}^{H \times W \times C}$, where $H$ is height, $W$ is width, and $C$ is the number of spectral channels, with corresponding spatial and temporal metadata. This metadata includes a *coordinate reference system* (CRS) that maps pixel coordinates to the surface of the Earth, a *spatial resolution* (the size of each pixel when mapped onto the surface of the Earth), *spatial bounds* (a bounding box representing the area on Earth that the data covers), and a *timestamp* or time range to indicate when the data was collected. We say that two datasets, $X^1$ and $X^2$ are *pixel-aligned* if $X^1_{i,j}$ and $X^2_{i,j}$ represent data from the same positions on Earth for all $i,j$. Most pairs of datasets are not aligned by default. For example, $X^1$ and $X^2$ can be captured by two satellites in different orbits and will only have partially overlapping spatial bounds, or $X^1$ will be satellite imagery while $X^2$ will be from a dataset of labels with a different CRS (see Figure 2). However, deep learning model training requires pixel-aligned patches of imagery—i.e. smaller crops from large scenes. Most models are trained

with mini-batch gradient descent and require input tensors in the format $B \times H \times W \times C$ where $B$ is the number of samples in a mini-batch and $W$ and $H$ are constant over all samples in the batch. At a higher level, training semantic segmentation models requires pairs of pixel-aligned imagery and masks.

Aligning two datasets requires *reprojecting* the data from one in order to match the CRS of the other, *cropping* the data to the same spatial bounds, and *resampling* the data to correct for differences in resolution or to establish the same underlying pixel grid. Typically, these are performed as pre-processing steps using GIS software such as QGIS, ArcGIS, or tools provided by GDAL—see Section A.4 for an example of a GDAL command to align two data layers. This requires some level of domain knowledge to perform correctly and does not scale to large datasets as it requires creating duplicates of the layer to be aligned. Further, this approach still requires an implementation of a dataset or data loader that can sample patches from the pre-processed imagery.

In TorchGeo, we facilitate this process by performing the alignment logic on-the-fly to create pixel-aligned patches of data sampled from larger scenes. Specifically, we implement the alignment logic in custom PyTorch Dataset classes that are indexed in terms of spatiotemporal coordinates. Given a query in spatiotemporal coordinates, a desired destination CRS, and a desired spatial resolution, the custom dataset is responsible for returning the corresponding reprojected and resampled data for the query. We further implement geospatial data *samplers* that generate queries according to different criteria (e.g. randomly or in a regular grid pattern). See Section 3 for a discussion on the implementation of these.

As our datasets are indexed by spatiotemporal coordinates we can easily compose datasets that represent different data layers by specifying a valid area to sample from. For example, if we have two datasets, $D^1$ and $D^2$, we may want to sample data from the intersection of the layers, $D^1 \cap D^2$, if one layer is imagery and the other layer is labels, or the union of the layers, $D^1 \cup D^2$, if both layers are imagery. The latter is particularly powerful for use in applications like multimodal learning [4] and data fusion [68]. As we describe in the following section, we implement generic dataset classes for a variety of common remotely-sensed datasets (e.g. Landsat imagery) that can be composed in this way. This allows users of the library to create their own multimodal datasets without having to write custom code.

Most importantly, these abstractions that TorchGeo creates—geospatial datasets and samplers—can be combined with a standard PyTorch data loader class to produce fixed size batches of data to be transferred to the GPU and used in training or inference. See Listing 1 for a working example of TorchGeo code for setting up a functional data loader that uses Landsat and Cropland Data Layer (CDL) dataset implementations. Our approach trades off required storage space for data loading time compared to pre-processing all data layers, but, crucially, does not require knowledge of GIS tooling. We benchmark our implementations in Section 4.3.

## 3 IMPLEMENTATION

The implementation of TorchGeo follows the design of other PyTorch domain libraries to reduce the amount of new concepts that a user must learn to integrate it with their existing workflow. We split TorchGeo up into the following parts:

**Datasets** Our dataset implementations consist of both *benchmark datasets* that allow users to interface with common datasets used in RS literature and *generic datasets* that allow users to interface with common geospatial data layers such as Landsat or Sentinel-2 imagery. Either of these types of datasets can also be *geospatial datasets*, i.e. datasets that contain geospatial metadata and can be sampled as such. These are part of the core contribution of TorchGeo and we describe them further in the following section.

**Samplers** We implement samplers for indexing into any of our *geospatial datasets*. Our geospatial datasets are indexed by bounding boxes in spatiotemporal coordinates (as opposed to standard fixed-length datasets of images which are usually indexed by an integer). The samplers generate bounding boxes according to specific patterns: randomly across all scenes in a dataset, random batches from single scenes at a time, or in grid patterns over scenes. Different sampling patterns can be useful for different model training strategies, or for running model inference over datasets.

**Models** Most existing model implementations (e.g. in torchvision) are fixed to accept 3 channel inputs which are not compatible with multispectral imagery. We provide implementations (or wrappers around well-established implementations) of common deep learning model architectures with variable-sized inputs and pre-trained weights. For example, models that use all of the Sentinel-2

multispectral bands as inputs. We also implement architectures from recent geospatial ML work such as the Fully Convolutional Siamese Network [12].

**Transforms** Similar to existing model implementations, some existing deep learning packages do not support data augmentation methods for multi-spectral imagery. We provide wrappers for augmentations in the Kornia [45] library (which does support augmentations over arbitrary channels) and implement transforms specific to geospatial data.

**Trainers** Finally, we implement model training recipes using the PyTorch Lightning library [61]. These include both dataset-specific training code, and general training routines—for example, an implementation of the BYOL self-supervision method [23].

## 3.1 DATASETS

We organize datasets based on whether they are *generic datasets* or *benchmark datasets* and based on whether or not they contain geospatial metadata—i.e. are a *geospatial dataset*.

Benchmark datasets are datasets released by the community that consist of both inputs and target labels for a specific type of task (scene classification, semantic segmentation, instance segmentation, etc.). These may or may not also contain geospatial metadata that allows them to be joined with other sources of data. In our opinion, one of the strongest components of existing deep learning domain libraries is the way that they make the use of existing datasets trivial. We aim to replicate this and, for example, include options that let users automatically download the data for a corresponding dataset. Table 1 lists the set of benchmark datasets that TorchGeo currently supports.

| Dataset | Task | Source | # Samples | # Categories | Size (px) | Resolution (m) | Bands |
|---|---|---|---|---|---|---|---|
| ADVANCE [27] | C | Google Earth, Freesound | 5,075 | 13 | 512x512 | 0.5 | RGB |
| BigEarthNet [54] | C | Sentinel-1/2 | 590,326 | 19–43 | 120x120 | 10 | SAR, MSI |
| EuroSAT [25] | C | Sentinel-2 | 27,000 | 10 | 64x64 | 10 | MSI |
| PatternNet [70] | C | Google Earth | 30,400 | 38 | 256x256 | 0.06–5 | RGB |
| RESISC45 [10] | C | Google Earth | 31,500 | 45 | 256x256 | 0.2–30 | RGB |
| So2Sat [71] | C | Sentinel-1/2 | 400,673 | 17 | 32x32 | 10 | SAR, MSI |
| UC Merced [64] | C | USGS National Map | 21,000 | 21 | 256x256 | 0.3 | RGB |
| COWC [40] | C, R | CSUAV AFRL, ISPRS, LINZ, AGRC | 388,435 | 2 | 256x256 | 0.15 | RGB |
| Tropical Cyclone Wind Estimation [37] | R | GOES 8–16 | 108,110 | - | 256x256 | 4K–8K | MSI |
| SEN12MS [51] | S | Sentinel-1/2, MODIS | 180,662 | 33 | 256x256 | 10 | SAR, MSI |
| CV4A Kenya Crop Type [19] | S | Sentinel-2 | 4,688 | 7 | 3,035x2,016 | 10 | MSI |
| ETCI 2021 Flood Detection [41] | S | Sentinel-1 | 66,810 | 2 | 256x256 | 5–20 | SAR |
| GID-15 [57] | S | Gaofen-2 | 150 | 15 | 6,800x7,200 | 3 | RGB |
| LandCover.ai [5] | S | Aerial | 10,674 | 5 | 512x512 | 0.25–0.5 | RGB |
| Smallholder Cashew Plantations in Benin [30] | S | Airbus Pléiades | 70 | 6 | 1,186x1,122 | 0.5 | MSI |
| Potsdam [49] | S | Aerial | 38 | 6 | 6,000x6,000 | 0.05 | MSI |
| Vaihingen [49] | S | Aerial | 33 | 6 | 1,281–3,816 | 0.09 | RGB |
| NWPU VHR-10 [9] | I | Google Earth, Vaihingen | 800 | 10 | 358–1,728 | 0.08–2 | RGB |
| SpaceNet 1, 2, 4, 7 [59] | I | WorldView-2/3, Planet Lab Dove | 1,889–28,728 | 2 | 102–900 | 0.5–4 | MSI |
| ZueriCrop [58] | I, T | Sentinel-2 | 116K | 48 | 24x24 | 10 | MSI |
| Seasonal Contrast [35] | T | Sentinel-2 | 100K–1M | - | 264x264 | 10 | MSI |
| LEVIR-CD+ [52] | D | Google Earth | 985 | 2 | 1,024x1,024 | 0.5 | RGB |
| OSCD [13] | D | Sentinel-2 | 24 | 2 | 40–1,180 | 60 | MSI |
| xView2 [24] | D | Maxar | 3,732 | 4 | 1,024x1,024 | 0.8 | RGB |

C = classification, R = regression, S = semantic segmentation, I = instance segmentation, T = time series, D = change detection

Table 1: Benchmark datasets implemented in TorchGeo.

Generic datasets are not created with a specific task in mind, but instead represent layers of geospatial data that can be used for any purpose. For example, we implement datasets for representing collections of scenes of Landsat imagery that let users index into the imagery and use it in arbitrary PyTorch-based pipelines. These are not limited to imagery, for example we also implement a dataset representing the Cropland Data Layer labels, an annual US wide raster layer that gives the estimated crop type or land cover at a 30 m/px resolution (see Table 2).

| Type | Dataset |
|---|---|
| Image Source | Landsat [50] |
| | Sentinel [16] |
| | National Agriculture |
| | Imagery Program [20] |
| Labels | Cropland Data Layer [7] |
| | Chesapeake Land Cover [46] |
| | Canadian Buildings Footprints [55] |

Table 2: Generic datasets implemented in TorchGeo.

## 3.2 SAMPLERS

As our *geospatial datasets* are indexed with bounding boxes using spatiotemporal coordinates and do not have a concept of a dataset "length", they cannot be sampled from by choosing a random integer. We provide three types of samplers for different situations: a "RandomGeoSampler" that returns a fixed-sized bounding box from the valid spatial extent of a dataset uniformly at random, a "RandomBatchGeoSampler" that returns a set of randomly positioned fixed-sized bounding boxes from a random scene within a dataset, and a "GridGeoSampler" that returns bounding boxes in a grid pattern over subsequent scenes within a dataset. These samplers are benchmarked in Section 4.2. This abstraction also allows for methods that rely on specific data sampling patterns. For example, Tile2Vec [29] relies on sampling triplets of imagery where two of the images are close to each other in space while the third is distant. This logic can be implemented in several lines of code as a custom sampler class, that would then operate over any of the generic imagery datasets. Finally, all TorchGeo samplers are compatible with PyTorch data loader objects so can be fit into any PyTorch-based pipeline.

## 4 EXPERIMENTS AND RESULTS

### 4.1 DATASETS

We use the following datasets in our experiments:

**Landsat and Cropland Data Layer (CDL)** The Landsat and CDL dataset consists of multispectral imagery from 114 Landsat 8 [50] collection 2 level 2 scenes taken in 2019 that intersect with the continental United States and the 2019 Cropland Data Layer (CDL) dataset [7]. This data is 151 GB on disk and is stored in cloud optimized GeoTIFF (COG) format. We use this dataset to benchmark our proposed GeoDataset and sampler implementations.

**So2Sat** The So2Sat-LCZ42 dataset [71] is a *classification* dataset that consists of 400,673 image patches classified with one of 42 different local climate zone labels. The patches are $30 \times 30$ pixels in size with 18 channels consisting of Sentinel 1 and Sentinel 2 bands. They are sampled from different urban areas around the globe. We use the second version of the dataset as described on the project's GitHub page[1] in which the training split consists of data from 42 cities around the world, the validation split consists of the western half of 10 other cities, and the testing split covers the eastern half of the 10 remaining cities.

**LandCover.ai** The LandCover.ai dataset is a *semantic segmentation* dataset [5] that consists of high-resolution (0.5 m/px and 0.25 m/px) RGB aerial imagery from 41 tiles over Poland where each pixel has been classified as one of five classes (four land cover classes and a "background" class). The scenes are divided into 10,674 $512 \times 512$ pixel patches and partitioned into pre-defined training, validation, and test splits according to the script on the dataset webpage[2].

**Chesapeake Land Cover** The Chesapeake Land Cover dataset [46] is a *semantic segmentation* dataset that consists of high-resolution (1 m/px) imagery from the US Department of Agriculture's National Agriculture Imagery Program (NAIP) and high-resolution (1 m/px) 6-class land cover

---

[1]https://github.com/zhu-xlab/So2Sat-LCZ42
[2]https://landcover.ai

labels from the Chesapeake Conservancy. Specifically, the dataset contains imagery and land cover masks for parts of six states in the Northeastern US: Maryland, Delaware, Virginia, West Virginia, Pensylvania, and New York. The data for each state is split into ∼7×6 km tiles and then divided into pre-defined train, validation, and test splits[3].

**RESISC45**  The RESISC45 dataset is a *classification* dataset [69] that consists of 31,500 256×256 pixel RGB image patches of varying spatial resolutions where each patch is classified into one of 45 classes. As the dataset does not have official splits, we use the train/val/test splits defined in [42].

**ETCI 2021**  The ETCI 2021 dataset [41] is a *semantic segmentation* dataset used in a flood detection competition. It consists of 66,810 256x256 pixel Sentinel-1 SAR images. We use the official train/test splits, and we further randomly subdivide the train split 80/20 into train/val splits.

**EuroSAT**  The EuroSAT dataset [25] is a *classification* dataset consisting of 27,000 64x64 pixel Sentinel-2 images and 10 target classes. As the dataset does not have official splits, we use the train/val/test splits defined in [42].

**UC Merced**  The UC Merced dataset [64] is a *classification* dataset consisting of 21,000 256x256 pixel RGB images from the USGS National Map and 21 target classes. As the dataset does not have official splits, we use the train/val/test splits defined in [42].

## 4.2 DATA LOADER BENCHMARKS

We first benchmark the speed at which TorchGeo can sample patches of imagery and masks from the Landsat and CDL dataset. We believe this dataset is typical of a large class of geospatial machine learning problems—where users have access to a large amount of satellite imagery scenes covering a broad spatial extent and, separately, per-pixel label masks where each scene is not necessarily projected in the same coordinate reference system. The end goal of such a problem is to train a model with pixel-aligned patches of imagery and label masks as described in Section 2. As such, we measure the rate at which our dataset and sampler implementations can provide patches to a GPU for training and inference.

In Figure 3a, we calculate the rate at which samples of varying batch size can be drawn from a "GeoDataset" using various "GeoSampler" implementations. Compared to the other samplers, "GridGeoSampler" is significantly faster due to the repeated access of samples that are already in GDAL's least recently used (LRU) cache. For small batch sizes, "RandomGeoSampler" and "RandomBatchGeoSampler" are almost identical, since overlap between patches is uncommon. However, for larger batch sizes, "RandomBatchGeoSampler" starts to outperform "RandomGeoSampler" as the cache is used more effectively.

In Figure 3b, we demonstrate the difference that preprocessing and caching data makes. This is most easily demonstrated by "GridGeoSampler" and to a lesser extent the other samplers. GDAL's LRU cache only saves raw data loading times, so warping must always be done on-the-fly if the dataset CRSs or resolutions

---

[3]https://lila.science/datasets/chesapeakelandcover

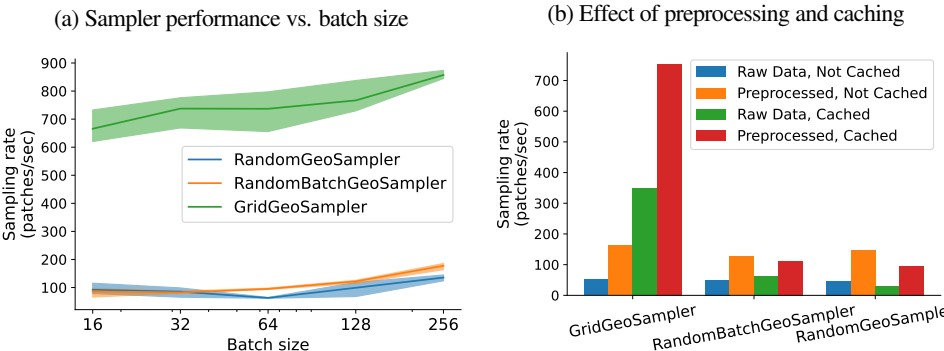

Figure 3: Sampling performance of various "GeoSampler" implementations. (a) Solid lines represent average sampling rate, while shaded region represents minimum and maximum performance across random seeds. (b) Average sampling rate under different data loading conditions.

don't match. When the necessary storage is available, preprocessing the data ahead of time can lead to significantly faster sampling rates. Although "RandomGeoSampler" and "RandomBatchGeoSampler" are much slower than "GridGeoSampler", most users will only need to use "GridGeoSampler" for inference due to our pre-trained model weights.

## 4.3 DATASET BENCHMARKS

We also use TorchGeo to create simple, reproducible benchmark results on 7 of the datasets described in Section 4.1. We report the uncertainty in the results such that future work can evaluate whether a proposed improvement is due to methodological innovation or variance in results due to the stochastic nature of training deep learning models. To ensure reproducibility, we include a model training and evaluation framework in TorchGeo based around the PyTorch Lightning library and release all of our results. To quantify uncertainty we report the mean and standard deviation metrics calculated over 10 training runs with different random seeds. The torchmetrics library [6] was used to compute all performance values (overall top-1 accuracy for the classification datasets, and mean intersection over union for the semantic segmentation datasets).

Our main results are shown in Table 3. We find that our simple training setup achieves competitive results on several datasets. The in-domain pre-training method in [42] trains models, starting with ImageNet weights, further on remote sensing (in domain) datasets before actually training on the target dataset and finds that this performs better than simply starting from ImageNet weights. In contrast, our best result across the RESISC45, EuroSAT, and UC Merced datasets come from simply using ImageNet pre-trained models and training on the target dataset with a low learning rate. The difference between these approaches is likely entirely due to different learning rate selection in the hyperparameter search, and data augmentation in the in-domain pre-training setup. On the So2Sat dataset we find that the ImageNet pre-trained models are able to achieve similar results as in-domain pretraining, but only when using all Sentinel-2 bands. The previously

| Dataset | Method | Weight Initialization | Bands | Performance |
|---|---|---|---|---|
| RESISC45 [10] | ResNet50 | ImageNet | RGB | $95.42 \pm 0.23\%$ |
| | ResNet18 | random | RGB | $79.90 \pm 0.25\%$ |
| | ResNet50 v2 [42] | In domain | RGB | **96.86%** |
| | ViT B/16 [53] | ImageNet-21k | RGB | 96.80% |
| | ResNet50 [66] | Sup-Rotation-100% | RGB | 96.30% |
| So2Sat [71] | ResNet50 | ImageNet (+ random) | MSI | **$63.99 \pm 1.38\%$** |
| | ResNet50 | random | MSI | $56.82 \pm 4.32\%$ |
| | ResNet50 | ImageNet | RGB | $59.82 \pm 0.94\%$ |
| | ResNet50 | random | RGB | $49.46 \pm 2.67\%$ |
| | ResNeXt + CBAM [71] | random | MSI | 61% |
| | ResNet50 v2 [42] | In domain | RGB | **63.25%** |
| LandCover.ai [5] | U-Net, ResNet50 encoder | ImageNet | RGB | $84.81 \pm 00.21\%$ |
| | U-Net, ResNet50 encoder | random | RGB | $79.73 \pm 00.67\%$ |
| | DeepLabv3+, Xception71 with DPC encoder [5] | Cityscapes | RGB | **85.56%** |
| Chesapeake Land Cover [46] | U-Net, ResNet50 encoder | ImageNet (+ random) | MSI | **$69.40 \pm 1.39\%$** |
| Delaware split | U-Net, ResNet18 encoder | random | MSI | **$68.99 \pm 0.84\%$** |
| ETCI 2021 [41] | U-Net, ResNet50 encoder | random | SAR | **$45.77 \pm 3.19\%$** |
| EuroSAT [25] | ResNet50 | ImageNet (+ random) | MSI | $97.86 \pm 0.23\%$ |
| | ResNet50 | random | MSI | $96.07 \pm 0.28\%$ |
| | ResNet50 | ImageNet | RGB | $98.11 \pm 0.31\%$ |
| | ResNet50 | random | RGB | $87.33 \pm 0.76\%$ |
| | ResNet50 v2 [42] | In domain | RGB | **99.20%** |
| UC Merced [64] | ResNet50 | ImageNet | RGB | $98.15\% \pm 0.46\%$ |
| | ResNet50 v2 [42] | In domain | RGB | **99.61%** |

Table 3: Benchmark results comparing TorchGeo trained models to previously reported results over seven datasets. RESISC45, So2Sat, EuroSat, and UC Merced results are reported as overall top-1 accuracy while LandCover.ai, Chesapeake Land Cover, and ETCI 2021 results are reported as mean class IoU. Results from TorchGeo models are reported as the mean with one standard deviation over 10 *training* runs from different random seeds. Results from related work are reported as is.

reported baseline methods on the LandCover.ai dataset all use a DeepLabV3+ segmentation model with a Xception71 + Dense Prediction Cell (DPC) encoder that has been pre-trained on Cityscapes. We are able to achieve a result within 0.75% mIoU of this setup using a simple U-Net and ResNet50 encoder pre-trained on ImageNet. Finally, we report the first set of basic benchmark results on the Chesapeake Land Cover dataset.

## 4.4 Effect of ImageNet pre-training on generalization performance

Two of the datasets we test with, So2Sat and Chesapeake Land Cover, contain splits that are designed to measure the generalization performance of a model. The validation and test splits from So2Sat include data from urban areas that are not included in the training split while the Chesapeake Land Cover dataset contains separate splits for six different states. In these setting we observe a large performance boost when training models from ImageNet weights versus from a random initialization, however we do not observe the boost on in-domain data. Table 4 shows the performance of models that are trained on the Delaware split and evaluated on the test splits from every state. The in-domain performance (i.e. in Delaware) of the ImageNet pre-trained model and model trained from scratch are the same, however in every other setting the ImageNet pre-trained model performs better. In all cases but Maryland, this difference is greater than 6 points of mIoU with the most extreme difference being 12 points in Virginia. In the So2Sat case we find that most models are not able to significantly reduce validation loss (where in this case validation data is out-of-domain), while, unsurprisingly, achieving near perfect results on the training data. Despite this overfitting, there remains a large gap between the best and worst models, with ImageNet pre-trained models achieving +7% and +10% accuracy over randomly initialized models. These results extend existing lines of research in computer vision that show how pre-training can improve out-of-domain performance [26], and how, specifically, ImageNet pretraining is useful for transfer learning tasks [28].

| Weight init | Delaware | Maryland | New York | Pennsylvania | Virginia | West Virginia |
|---|---|---|---|---|---|---|
| ImageNet (+ random) | 69.40 ± 1.39% | 59.57 ± 0.70% | 57.95 ± 1.10% | 55.13 ± 1.25% | 45.56 ± 1.54% | 20.76 ± 1.95% |
| random | 68.99 ± 0.84% | 57.30 ± 0.78% | 49.26 ± 2.40% | 47.67 ± 2.40% | 33.14 ± 3.73% | 14.95 ± 2.72% |

Table 4: Mean IoU performance of models trained in Delaware, with and without ImageNet weight initialization, on the test splits from Chesapeake Land Cover dataset.

## 5 Discussion

We introduce TorchGeo, a Python package for enabling deep learning with geospatial data. TorchGeo provides data loaders for common geospatial datasets, composable data loaders for arbitrary geospatial raster and vector data, samplers appropriate for geospatial data, models, transforms, and model trainers. Importantly, TorchGeo allows users to bypass the common pre-processing steps necessary to align geospatial data with imagery and performs this processing on-the-fly. We benchmark TorchGeo dataloader speed, and demonstrate how TorchGeo can be used to create reproducible benchmark results in several geospatial datasets.

Finally, TorchGeo serves as a platform for performing geospatial machine learning research. Existing works in self-supervised learning with geospatial data rely on spatiotemporal metadata and can be naturally implemented in TorchGeo and scaled over large amounts of geospatial imagery without the need for pre-processing steps. Similarly, augmentation methods appropriate for training with geospatial imagery are under-explored, however can be integrated easily with TorchGeo. Other interesting directions include building inductive biases appropriate for geospatial imagery into deep learning models (similar to the work done on rotation equivariant networks [36]), data fusion techniques (e.g. how to incorporate spatial information into models, or appropriately use multi-modal layers), and learning shape-based models. Finally, TorchGeo exposes a catalog of benchmark geospatial datasets (Table 1) through a common interface, and, with the results in this paper, has begun to include corresponding benchmark results. This makes it easy for researchers to compare new ideas to existing work without having to repeat expensive computations. We hope TorchGeo can help drive advances at the intersection of machine learning and remote sensing.

ETHICS STATEMENT

Machine learning with remotely sensed data has the potential to impact many aspects of society in both positive and negative ways. For example, satellite imagery can be used to model and track progress towards roughly 20% of the indicators of the United Nation's sustainable development goals [2], as well as inform more effective responses to natural disasters. At the same time, remotely sensed imagery can be used for unwarranted tracking of individuals. For example, academic papers and datasets about human and vehicle re-identification from high-resolution imagery exist and pose serious ethical concerns. Further, the biases of models that are run over satellite imagery are not well understood, and, if used in decision making, could systematically exacerbate existing disparities. TorchGeo makes it easier for researchers to develop machine learning models with remotely sensed data. Users of TorchGeo, and the wider machine learning and remote sensing communities, should consider the ethical consequences of their research when publishing work.

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

# A APPENDIX

## A.1 DATA LOADER BENCHMARKING

In order to benchmark the performance of our data loaders, we download 114 Landsat 8 scenes and 1 Cropland Data Layer (CDL) file for the year of 2019. All files are stored as Cloud Optimized GeoTIFFs (COGs) and kept in their original CRS (Albers Equal Area for CDL and UTM for Landsat). Files are uncompressed with a block size of 512. Experiments are run on Microsoft Azure with a 6-core Intel Xeon E5-2690 CPU. All data is stored on a local SSD attached to the compute node. Batch size and random seed are varied while the remaining hyperparameters are kept fixed. Total epoch size is 4096, patch size is 224, stride is 112, and number of workers for parallel data loading is set to 6.

## A.2 BENCHMARK DATASET EXPERIMENTS

For experiments, we use the pre-defined training, validation, and testing splits in all datasets. We perform a small hyperparameter search with a single fixed random seed on each dataset. Specifically we search over the best validation performance over the grid: learning rate $\in \{0.01, 0.001, 0.0001\}$, loss function $\in \{$cross entropy, jaccard$\}$, weight initialization $\in \{$random, ImageNet$\}$ and model architecture $\in \{$ResNet18, ResNet50$\}^4$. With the So2Sat and EuroSAT datasets we also run experiments that use all the Sentinel 2 bands vs. only the RGB bands. In the cases where we use ImageNet weights with imagery

---

[4]For the classification problems we use the ResNets *as is* and for the semantic segmentation datasets we use the ResNets as the encoder model in a U-Net.

that has more than RGB bands we randomly initialize the non-RGB kernels in the first convolutional layer of the network and denote this setting as ImageNet (+ random). In all cases we use the AdamW optimizer [33], reduce learning rate on validation loss plateaus, and early stop based on validation loss. We repeat the training process with 10 different seeds using the best performing hyperparameter configuration and report the test set performance over these models.

## A.3 TORCHGEO CODE EXAMPLE

TorchGeo is designed to be simple and easy to use, and provides a familiar API for users who have experience using libraries like torchvision [38]. In Listing 1, we provide an example code snippet showing how to use TorchGeo. In this example, we first instantiate dataset objects for Landsat 7 and 8 [50]. We then take the union of these two datasets, treating them as equivalent such that it doesn't matter which satellite the imagery comes from. Next we instantiate a Cropland Data Layer (CDL) [7] dataset. We perform an intersection between our Landsat and CDL datasets so that we sample only from the regions of overlap between the two. We also instantiate a PyTorch sampler, allowing us to sample from this dataset using spatiotemporal queries. All use of latitude/longitude bounding boxes, coordinate reference systems (CRS), and spatial resolution are abstracted away from the user, allowing non-remote sensing experts to work effectively with geospatial data. All datasets and samplers are compatible with PyTorch's DataLoader class, making it easy to modify an existing PyTorch workflow to handle geospatial data.

```python
from torch.utils.data import DataLoader
from torchgeo.datasets import Landsat7, Landsat8, CDL
from torchgeo.samplers import RandomGeoSampler

# Take the union of all Landsat 7 and 8 imagery
landsat7 = Landsat7(root="...")
landsat8 = Landsat8(root="...")
landsat = landsat7 | landsat8

# Take the intersection of Landsat and CDL data
cdl = CDL(root="...")
dataset = landsat & cdl

# Sample 1 km^2 image patches from the intersection of the datasets
sampler = RandomGeoSampler(dataset, size=1000, length=512)

# Use the dataset and sampler as normal in a PyTorch DataLoader
dataloader = DataLoader(dataset, sampler=sampler, batch_size=32)
for batch in dataloader:
    # Train a model, or make predictions using a pre-trained model
```

Listing 1: Example TorchGeo code for creating a joint Landsat 7/8 [50] and Cropland Data Layer (CDL) [7] dataset, and using such a dataset with a standard PyTorch DataLoader class.

## A.4 PRE-PROCESSING ALIGNMENT WITH GDAL

As an example of an alignment pre-processing workflow, we assume that we have a Landsat 8 scene and a Cropland Data Layer (CDL) raster ("cdl.tif") which completely covers the extent of the Landsat scene. We would like to create a pixel-aligned version of these two layers. Given that the Landsat 8 scene has a CRS of "EPSG:32619", a height of 8011 pixels, a width of 7891 pixels, and spatial bounds of (186585, 4505085, 423315, 4745415), the corresponding GDAL command would create a cropped version of the CDL layer that is aligned to the Landsat layer:

```
gdalwarp \
    -t_srs EPSG:32619 \
```

```
-of COG \
-te 186585 4505085 423315 4745415 \
-ts 7891 8011 \
cdl.tif aligned_cdl.tif
```

The spatial metadata of the Landsat scene can be determined through other GDAL command-line tools (gdalinfo command), geospatial data packages such as the "rasterio" package in Python, or through GIS software such as QGIS or ArcGIS.

