# OpenReview forum: "TorchGeo: deep learning with geospatial data"
_ICLR.cc/2022/Conference — ICLR 2022 Submitted_

### Official Review · Reviewer_6sDp · 2021-10-20

**Correctness:** 4
**Technical Novelty And Significance:** 2
**Empirical Novelty And Significance:** 2
**Recommendation:** 5
**Confidence:** 3

**Main Review:**

Strengths : The authors implement various architecture combined with popular datasets

Weaknesses: The technical novelty of the paper is low (mathematical modeling, deep learning architecture, processing, etc..)

**Summary Of The Paper:**

The paper introduces TorchGeo, a Python library for integrating geospatial data into the PyTorch deep learning ecosystem. TorchGeo provides data loaders for a variety of benchmark datasets, composable datasets for generic geospatial data sources, samplers for geospatial data, and transforms that work with multispectral imagery

**Summary Of The Review:**

Even though the library is interesting, the overall contribution of the paper is low and does not advance the field

---

> ### Author Response · Authors · 2021-11-16
> **Rebuttal**
>
> Thank you for the review. Our paper contains methodological novelty, important/interesting results, and describes a project with larger implications for the machine learning and remote sensing communities. It fits within the scope of the conference per the [ICLR call for papers](https://iclr.cc/Conferences/2022/CallForPapers). Please see the response to the other reviewers for an expanded argument about the novelty/fit, new results, and other improvements we’ve made. Again, thank you for your time.

---

### Official Review · Reviewer_W3nK · 2021-10-24

**Correctness:** 3
**Technical Novelty And Significance:** 3
**Empirical Novelty And Significance:** 3
**Recommendation:** 6
**Confidence:** 4

**Main Review:**

- Establishes an easily accessible and optimized pipeline for data processing that is often esoteirc
- A wide range of datasets are available

Weakness:
- Can detail the optimization (CPU vs GPU) process, availability and ease of adding new metrics, plots, figures etc.
- Give examples of tasks
- Detail how this pipeline will actually be used, eg reduced time, reduced lines of code etc
- Encourage people to add their own datasets to this library, and detail the process for doing so
- Have datasheets for datasets also published

**Summary Of The Paper:**

Optimized Data downloading, processing, and sampling especially for domain scientists can be cumbersome. These are often steps that are routinely taken and can be optimized well once so that the entire community benefits from the ML-readiness. This paper tries to create generalized pipelines that both domain scientists and AI researchers can use to quickly get up and running with their experiments. The paper gathers existing datasets and makes them available through TorchGeo, a PyTorch domain library.

**Summary Of The Review:**

The paper establishes a new domain specific library that is ML-ready

---

> ### Author Response · Authors · 2021-11-16
> **Rebuttal**
>
> Thank you for your review and your interest in the library! We’ve noticed that most of the points you have listed as weaknesses are mainly technical questions about the library itself. Could you further clarify how you interpret these as weaknesses in the paper? (or, if we’ve sufficiently answered your questions below and you believe the paper belongs in ICLR, then please consider raising your score!)
>
> > Can detail the optimization (CPU vs GPU) process, availability and ease of adding new metrics, plots, figures etc.
>
> Most of the CPU/GPU optimization is handled by libraries like rasterio/GDAL or PyTorch/CUDA, not by TorchGeo itself. However, it's easy for users to miss crucial details like reprojection (see Figure 2) without some level of expertise in both geospatial data preprocessing and deep learning. The biggest contribution of TorchGeo is not making data loading or GPU processing more efficient, but to provide a standardized platform for handling it in the context of deep learning frameworks and abstracting the details away from users.
>
> > Give examples of tasks
>
> Can you elaborate on this? Do you mean remote sensing tasks (land cover mapping, change detection) or computer vision tasks (semantic segmentation, classification) or something else?
>
> > Detail how this pipeline will actually be used, eg reduced time, reduced lines of code etc
>
> Appendix A.2 gives an example of how simple the code is to load and merge two datasets in different coordinate systems and resolutions. This is in contrast with Appendix A.3, which would need to be run manually by the user on every file in the dataset, doubling the amount of storage used on disk.
>
> > Encourage people to add their own datasets to this library, and detail the process for doing so
>
> Once the paper is accepted and the double-blind period ends, we'll add a link to the GitHub repo and documentation for TorchGeo. We have already written extensive documentation on how to contribute to the library that covers topics like: writing unit tests, running style linters, and updating the documentation/tutorials. We don’t believe these types of details belong in the paper.
>
> > Have datasheets for datasets also published
>
> We don't yet provide standardized datasheets for all datasets, although, in addition to Table 1, our documentation does contain details on dataset image sources and classes and provide links to external documentation. This is something we will definitely consider for the future though!

---

### Official Review · Reviewer_Xe57 · 2021-10-31

**Correctness:** 3
**Technical Novelty And Significance:** 2
**Empirical Novelty And Significance:** 3
**Recommendation:** 5
**Confidence:** 4

**Details Of Ethics Concerns:**

Deep learning with GIS has a plethora of ethical concerns associated with it. However, I believe most of these are offset by the fact that we need these tools in order to best mitigate climate change. Providing an open source package that lowers the bar for computer-vision researchers to tackle problems in the geospatial domain, I think ultimately provides a net-positive effect from an perspective. As always, practitioners must remain alert and consider the ethical consequences when asked to apply these tools for specific tasks.

**Main Review:**

## Strengths


* Provides a good description of domain challenges when working with data indexed by geospatial coordinates.

* Provides ready-to-use torch datasets for 18 different benchmark geospatial datasets.

* Provides mean/std over 10 runs to quantify uncertainty in experimental metrics in 4 of these dataset.

* The torchgeo package abstracts away much of the geospatial domain knowledge and data-munging to make it easier for CV practitioners to experiment on geospatial datasets.

* The question of if the inductive biases from ImageNet will transfer to geospatial domains is interesting and a question I've had myself. I think the results in this section are generally interesting wrt to how well ImageNet generalizes to other domains.



## Weaknesses

* I's suggest using a different acronym than EO when talking about Earth Observation. Perhaps just remote sensing (RS) is better? Working within this domain, I find EO is more strongly associated with Electro-Optical than Earth Observation, but this is just a suggestion and not a big deal.

* The abstract claims it is is the first package to publish pretrained models with MSI imagery, but there are many variants of MSI imagery, so I think it is important to say what bands are used. I assume these are the Sentinel 2 bands? I think it would be sufficient to say that.

* Indexing in terms of geo-spatial coordinates still requires domain knowledge. IMO a more natural (wrt to a computer-vision researcher) alternative method might be to index by pixels within a geospatial region at a given GSD (pixel resolution). But there are certainly advantages to the former. The point is that it is still domain knowledge.

* Torch datasets return both data and target label, the choice of data and label will depend heavily on the task to be learned (e.g. patch category labels, segmentations). While this is fine for providing data loaders for pre-defined datasets, this does not provide a way to ingest custom geo-spatial datasets. This may be out of the scope of this paper, so I'm not sure how big of a weakness this is.

* Benefit of a torch-specific library are by design restricted only to torch users.

* Many of the challenges dealing with geospatial data (large image sizes, multiple input bands) are not unique to the geospatial domain. The medical domain has similar challenges, but the tooling for these domains is often developed independently without considering common core infrastructure issues that could be jointly addressed (e.g. openslide vs cog formats).

* It is unclear what "cached" means in Figure 3. I assume it's used wrt to GDAL's cache? I use GDAL heavily, but I'm unfamiliar with how it's internal cache works, but I would expect that it's some sort of LRU cache. In that case it's not clear how cached versus not cached is measured. The other possibility I considered is that perhaps the raw data is not stored as a COG, so there is some cache that creates a tile-based format for efficient patch sampling. That would make more sense to me wrt to this cache, but again it's unclear and I would encourage the authors to clarify here.

* I disagree strong with the idea from Section 4.3 that controlling for the pseudorandom number generator is necessary to determine if benefits are caused by design decisions or just getting a good seed. This would imply that we should never train with non-deterministic algorithms like those employed in CUDNN. Instead I think it underscores the importance of gathering statistics with quantified uncertainty bounds about a method over multiple runs before determining if there is an improvement or not. If a PRNG is causing a spike in performance, you likely have a sharp-non-generalizing model.

* The model evaluation metrics used are unclear. If this paper is going to claim reproducible results, I would like to know what software is used to calculate quality metrics. Are metrics baked into torchgeo? Is the pycocoutils implementation used? Is it something else?

* I don't think you can claim SotA on RESISC45 if you don't have the same train/validation/test split as the original paper. Have you contacted the authors to try and get their split? Furthermore, you should ensure that you publish your train/val/test split in torchgeo so that future work can compare to yours. If this is already done, mention it and say that you were unable to acquire the original split, so caveat any potential SotA claim with that, and then say that your split is published in torchgeo.

* The paper briefly mentions SAR data, but then seems to focus on mostly on RGB data in experiments, with the exception being So2Sat with includes some MSI data. There is no discussion of how augmentations might differ from RGB augmentations when applied to MSI or SAR, or at least what examples of some augmentations appropriate or not appropriate for each might be.

* In the appendix I would like to know the COG settings used in this package. If data is not
  provided in COG or some tiled format, is it cached as a COG? If so what
compression and blocksize settings are used?

* In the appendix it's unclear how I would go about merging MSI datasets with different bands / resolutions. If I was loading and merging L8 and an S2 dataset, what would happen?

* How does pixel alignment work when faced with Maxar imagery that may require RPC transformations with a digital elevation model in order to get accurate geo-localization?  Does torch-geo take that into account? Does it attempt to always return orthorectified data?

* Does torch geo do anything to align images over time, say for change-detection datasets like XView2? Can I ask a dataset for all patches in a region within a certain time frame?

* What are the license restrictions on the datasets that torch geo provides loaders for? Does it automatically download the datasets like torchvision does? I imagine this is why some datasets are not supported (e.g. xview datasets)?

* It's unclear to me what the backend used to maintain the datasets and annotations are. Are they different for each loader or does torchgeo use a common data format to index all of the datasets.

* At the end of section 2 it says the approach trades off space for time, and I can guess at what that means, but I think the authors should state it more explicitly.

* The experiments in 4.2 show the effect of different dataloaders, but does not compare against any baseline naive method of sampling. It might help to underscore the importance of domain-aware dataloading methods if you compared to something like a torchvision ImageFolder dataset. I imagine it's lack of ability to work with tiled formats like COGs would clearly show the importance of your dataloaders.

**Summary Of The Paper:**

Introduces the torchgeo Python module, which provides torch Datasets, augmentations, samplers, and pretrained models to provide reproducible data ingestion for deep learning on standard geospatial datasets.

These datasets are indexed in terms of geo-spatial coordinates, which make it natural to express the desired sampling pattern.

The paper investigates the question of how well ImageNet RGB models transfer to multispectral geospatial models.

The paper investigates the efficiency of different sampling strategies and provides experimental measurements.

Trains baseline models and achieves competitive or state-of-the-art results for 4 datasets considered.

**Summary Of The Review:**

I work with geospatial data, and this paper does a good job at describing the challenges, and I think it's important that packages like this are developed and publicized. The paper doesn't introduce anything particularly novel, but I think the goal of providing reproducible methods for experimentation is technically significant enough to warrant publication.

There are a few issues in the weaknesses that I would like to see resolved, but if these are addressed I think this is a above the acceptance threshold. Particularly what I'd like to see is more discussion of how torchgeo handles MSI/SAR data and how (or if) these disparate modalities can be rectified and combined when merging different datasets.

Lastly, I'd like to mention that I've done a lot of work in this domain, and after the review period is over I'd be interested in connecting with the authors to share ideas and software in order to help make this package as generally useful as possible.

---

> ### Author Response · Authors · 2021-11-16
> **Rebuttal**
>
> Thanks for the detailed feedback, it is extremely helpful! We would love to get in touch after the review period is over. Our library is open source, and we will include a link to the repository after the review period ends.
>
> To address the weaknesses you pointed out, we'll go through each bullet point one at a time:
>
> > I's suggest using a different acronym than EO when talking about Earth Observation. Perhaps just remote sensing (RS) is better? Working within this domain, I find EO is more strongly associated with Electro-Optical than Earth Observation, but this is just a suggestion and not a big deal.
>
> We agree that this can be confusing, and have replaced all usage of EO with RS.
>
> > The abstract claims it is the first package to publish pretrained models with MSI imagery, but there are many variants of MSI imagery, so I think it is important to say what bands are used. I assume these are the Sentinel 2 bands? I think it would be sufficient to say that.
>
> So far, the pre-trained models we have available are for Sentinel-2 MSI imagery. We have now clarified this in the Models description in Section 3: Implementation.
>
> > Indexing in terms of geo-spatial coordinates still requires domain knowledge. IMO a more natural (wrt to a computer-vision researcher) alternative method might be to index by pixels within a geospatial region at a given GSD (pixel resolution). But there are certainly advantages to the former. The point is that it is still domain knowledge.
>
> We agree; while our datasets are indexed in geo-coordinates, we provide several types of custom “Sampler” classes that can be combined with a GeoDataset in a normal PyTorch Dataloader. This completely abstracts vision researchers away from geographic coordinates in most scenarios (they will need to decide whether they would like to randomly sample chips/patches from all data, sample chips/patches in a systematic gridded pattern, etc.).
> We have thought about other ways to accomplish this (e.g. having users specify an AOI, CRS, and GSD at the Dataset level—then using pixel coordinates) and would be happy to discuss further. The main issue with pixel-based indexing approaches is that they only work when all data is pixel-aligned, which is not generally the case for most datasets.
>
> > Torch datasets return both data and target label, the choice of data and label will depend heavily on the task to be learned (e.g. patch category labels, segmentations). While this is fine for providing data loaders for pre-defined datasets, this does not provide a way to ingest custom geo-spatial datasets. This may be out of the scope of this paper, so I'm not sure how big of a weakness this is.
>
> Torch datasets can return dictionaries with different keys. By convention, all of our builtin datasets return a dictionary with an “image” key and either a “label” or a “mask” key. The custom geospatial datasets can return either the “image” or “mask” key and can be composed with each other with different types of spatial joins. Specifically, we provide RasterDataset and VectorDataset base classes similar to torchvision's ImageFolder class that recursively find and load geospatial data files from a directory.
>
> We’ve also considered adding a “STACDataset” class that can be instantiated with a STAC, using something like the label extension to specify how different items should be treated, but haven't yet implemented this.
>
> > Benefit of a torch-specific library are by design restricted only to torch users.
>
> A fair point! We note that truly determined users can always use a PyTorch-based data loader, cast the data to a numpy array, and then pass it to a different ML/DL platform like TensorFlow. Also, as our library is open-source and distributed under the MIT license, future work on other platform specific geospatial ML libraries can take advantage of the organization/documentation that we have started.

---

> > ### Comment · Reviewer_Xe57 · 2021-11-30
> > **Response to rebuttal**
> >
> > Pending reviewing the updated paper, (which looks like it has still not been uploaded), these changes seem like they will be adequate enough for me to bump my recommendation. Looking forward to the updated pdf.

---

> > > ### Author Response · Authors · 2021-11-30
> > > **Response to response to rebuttal**
> > >
> > > We updated the paper most recently on the 23rd, is there something specific that we forgot to include?

---

> > > > ### Comment · Reviewer_Xe57 · 2021-12-01
> > > > **Re Response to response to rebuttal**
> > > >
> > > > Ah, I see it now.

---

> ### Author Response · Authors · 2021-11-16
> **Rebuttal 2**
>
> > Many of the challenges dealing with geospatial data (large image sizes, multiple input bands) are not unique to the geospatial domain. The medical domain has similar challenges, but the tooling for these domains is often developed independently without considering common core infrastructure issues that could be jointly addressed (e.g. openslide vs cog formats).
>
> We agree, and have made a point to contribute any library features that aren't specific to working with geospatial data in PyTorch to other libraries. For example, any data augmentation tools that involve multispectral imagery, but don't involve geospatial data, have been contributed to libraries like Kornia so that researchers in the medical domain can benefit from these. We try to keep this library as focused as possible partly to avoid duplicating functionality in multiple libraries (as you've mentioned) but also to reduce our own burden on having to maintain all of these features.
>
> > It is unclear what "cached" means in Figure 3. I assume it's used wrt to GDAL's cache? I use GDAL heavily, but I'm unfamiliar with how it's internal cache works, but I would expect that it's some sort of LRU cache. In that case it's not clear how cached versus not cached is measured. The other possibility I considered is that perhaps the raw data is not stored as a COG, so there is some cache that creates a tile-based format for efficient patch sampling. That would make more sense to me wrt to this cache, but again it's unclear and I would encourage the authors to clarify here.
>
> Yes, we are using GDAL's cache. GDAL uses an LRU cache whose size is controlled by the GDAL_CACHEMAX environment variable. The data we test on is stored in COG format (tiled) such that windowed reads only access one or two tiles at a time instead of loading the entire file into memory. If the same tile is accessed multiple times in a row (common with GridGeoSampler) then this cache is reused and data I/O is much faster.
>
> > I disagree strong with the idea from Section 4.3 that controlling for the pseudorandom number generator is necessary to determine if benefits are caused by design decisions or just getting a good seed. This would imply that we should never train with non-deterministic algorithms like those employed in CUDNN. Instead I think it underscores the importance of gathering statistics with quantified uncertainty bounds about a method over multiple runs before determining if there is an improvement or not. If a PRNG is causing a spike in performance, you likely have a sharp-non-generalizing model.
>
> We have rewritten parts of 4.3 to clarify what we mean. Training a deep learning network is a stochastic process where the final result for a specific training run depends on the PRNG seed used. As such, we think that average results (over the randomness of initialization and mini-batches) with a measure of spread should be reported in all cases where comparisons between models (or training setups, or weight init methods, etc.) are being made. We’re not attempting to claim anything further than this.
>
> We note that some of the results we report in Table 3 do have significant variance between random seeds. E.g. models trained on So2Sat from _the same_ ImageNet weight initialization (and other hyperparameters) have results with a standard deviation of 1.38% top-1 accuracy simply due to the PRNG seed used at the beginning of the training process. I.e. the 10 training runs converge to different local minima with different performance on the test set. The size of this difference varies by dataset, but we often find results in related work that are reported without any error bounds/confidence intervals (e.g. none of the papers we cite results from in Table 3 estimate this).
>
> > The model evaluation metrics used are unclear. If this paper is going to claim reproducible results, I would like to know what software is used to calculate quality metrics. Are metrics baked into torchgeo? Is the pycocoutils implementation used? Is it something else?
>
> This is a good point. All metrics were computed using the torchmetrics library and this has now been clarified in the paper. We are reminded of work such as [1] in which the calculation of FID differs based on bugs in various library implementations.
>
> [1] Parmar, Gaurav, Richard Zhang, and Jun-Yan Zhu. "On Buggy Resizing Libraries and Surprising Subtleties in FID Calculation." arXiv preprint arXiv:2104.11222 (2021). https://arxiv.org/pdf/2104.11222.pdf

---

> ### Author Response · Authors · 2021-11-16
> **Rebuttal 3**
>
> > I don't think you can claim SotA on RESISC45 if you don't have the same train/validation/test split as the original paper. Have you contacted the authors to try and get their split? Furthermore, you should ensure that you publish your train/val/test split in torchgeo so that future work can compare to yours. If this is already done, mention it and say that you were unable to acquire the original split, so caveat any potential SotA claim with that, and then say that your split is published in torchgeo.
>
> Thanks for catching this. We re-tested with the splits used in the “In-domain representation learning for remote sensing” paper and indeed do not have SOTA results. We’ve updated our results and the paper appropriately and changed the RESISC45 dataset implementation in torchgeo to use these splits as the default for future work to build off of.
>
> Further, we looked into the difference in performance between our initially reported results and the new results. While the RESISC45 dataset is balanced (and our initial results were based off of a random balanced subset), the splits used in the “In-domain…” paper are chosen randomly (i.e. do not preserve the class balance). E.g. the most common class in the training set has 460 samples while the least frequent has 391. Simply re-weighting the standard cross entropy loss function with the inverse of the training set class frequencies gives a boost to performance that roughly ties the best result from the In-Domain paper (we also found that standard augmentation methods help, however report unaugmented results in the paper).
>
> > The paper briefly mentions SAR data, but then seems to focus on mostly on RGB data in experiments, with the exception being So2Sat with includes some MSI data. There is no discussion of how augmentations might differ from RGB augmentations when applied to MSI or SAR, or at least what examples of some augmentations appropriate or not appropriate for each might be.
>
> We've added additional SAR and MSI datasets to our experimental results section, and clarified that most of the datasets we test on are SAR/MSI, not RGB-only. See the general response text.
>
> > In the appendix I would like to know the COG settings used in this package. If data is not provided in COG or some tiled format, is it cached as a COG? If so what compression and blocksize settings are used?
>
> We have added COG settings to the appendix.
>
> > In the appendix it's unclear how I would go about merging MSI datasets with different bands / resolutions. If I was loading and merging L8 and an S2 dataset, what would happen?
>
> Great question, we added an explanation of this to our general response and updated the paper to clarify this.
>
> > How does pixel alignment work when faced with Maxar imagery that may require RPC transformations with a digital elevation model in order to get accurate geo-localization? Does torch-geo take that into account? Does it attempt to always return orthorectified data?
>
> At the moment, the only type of on-the-fly preprocessing that is done is reprojection and resampling. More complicated on-the-fly pre-processing steps, like orthorectification or registration, are possible but would likely slow the data loaders down to the point of being useless (the geodatasets are already CPU bound). An interesting research/development direction that we have discussed is porting the resampling and reprojection steps to run on the GPU. However, currently, users must provide inputs to GeoDataset that have been preprocessed to a level appropriate for their application.

---

> ### Author Response · Authors · 2021-11-16
> **Rebuttal 4**
>
> > Does torch geo do anything to align images over time, say for change-detection datasets like XView2? Can I ask a dataset for all patches in a region within a certain time frame?
>
> You can define a custom GeoSampler that does exactly this. We'll likely be adding these kinds of samplers in the future as many self-supervised training approaches involve sampling the same location at different times. Also, agricultural applications have been shown to benefit from time series data like you propose.
>
> > What are the license restrictions on the datasets that torch geo provides loaders for? Does it automatically download the datasets like torchvision does? I imagine this is why some datasets are not supported (e.g. xview datasets)?
>
> Each dataset has its own licensing restrictions. Some datasets can be automatically downloaded (like with torchvision) while others (like xView2) require that users manually download the data. We do not attempt to redistribute datasets where the license forbids this, and users have to opt-in to dataset downloading. By the way, we've now added a dataset for xView2, thanks for the suggestion!
>
> > It's unclear to me what the backend used to maintain the datasets and annotations are. Are they different for each loader or does torchgeo use a common data format to index all of the datasets.
>
> All GeoDatasets use an R-tree to track the location of each image in time and space. To combine two different GeoDatasets, these R-trees must share the same CRS. For datasets that lack geospatial information, the dataset is much more similar to torchvision, where images are sampled by integer index.
>
> > At the end of section 2 it says the approach trades off space for time, and I can guess at what that means, but I think the authors should state it more explicitly.
>
> As a user, you either have to preprocess all of your data beforehand (which would require double the space on disk), or you have to preprocess your data on-the-fly (which would slow down data loading). This has been clarified in the paper.
>
> > The experiments in 4.2 show the effect of different dataloaders, but does not compare against any baseline naive method of sampling. It might help to underscore the importance of domain-aware dataloading methods if you compared to something like a torchvision ImageFolder dataset. I imagine it's lack of ability to work with tiled formats like COGs would clearly show the importance of your dataloaders.
>
> You make a good point, the main benefit of TorchGeo is not the speed with which you can load images, but the fact that you can load small windows of data from much larger COGs. No matter how much we optimize things, the large file size, warping, and resampling all add significant overhead, so data loading for geospatial data will never be as fast as for simple ImageFolder datasets.
>
> > Ethics
>
> Finally, we agree with the points you made about the ethical considerations of satellite imagery and machine learning. We think this is an important point to note, so have added an ethics statement (taking advice from the NeurIPS guidance) to the paper.

---

### Official Review · Reviewer_stxS · 2021-11-02

**Correctness:** 3
**Technical Novelty And Significance:** 3
**Empirical Novelty And Significance:** 4
**Recommendation:** 5
**Confidence:** 4

**Main Review:**

The paper is well written and has a good flow. The authors work upon a genuine issue of processing geo-spatial data with deep learning models. Geo-spatial datasets can be challenging compared to regular RGB imagery given the spectral bands, along with the embedded coordinate reference systems, that they have. The paper provides useful functions for indexing geo-spatial datasets on the basis of their bounding boxes, and pixel-aligning multi-sensor images, as well as data augmentations for multispectral images. The authors state that the library is also compatible with PyTorch dataloaders, which is good.

TorchGeo right now provides a set of benchmark datasets, consisting of MSI and RGB images. I assume the library would have the potential to be expanded later on to consist of hyperspectral and UAV imageries, along with Lidar as well as DEM data. Having said that, I believe providing benchmark geospatial datasets would be incomplete without providing access to some ISPRS benchmarks (https://www.isprs.org/education/benchmarks.aspx) such as Potsdam and Vaihingen datasets, especially in Section 4.1 and Table 3.

My major concern is what authors mention in the second last line of the Introduction, (also evident from Tables 3 & 4) - “ImageNet pretraining significantly improves spatial generalization performance” - why is this so? Why does ImageNet pre-training perform better than in-domain training? Is this solely because of the difference in chosen hyperparameters? A more detailed discussion on this observation from the perspective of vision (covering concepts such as image features, edges, scale of the ImageNet images vs remotely sensed images etc.) would be helpful to the community. Please expand Section 4.4 in this regard.

Table 2: Please give references to the labelled datasets (CDL, Chesapeak Land Cover etc.)

Table 3: The table does not provide pre-trained models on multispectral or SAR images. It only provides pre-trained models for mostly RGB imageries. As your contribution is towards geospatial datasets such as multispectral and SAR images, you should also provide pre-trained models on these datasets. Example: datasets presented in Table 1.

Section 3.2 and 4.2: Does any past work, such as in GDAL, also implement sampling methods? Can you compare the performance of your methods, their utility, as compared to the past methods?

Minor Comments:

 - Please cite Figure 1 and Table 2 in the main text
 - There’s no need to embolden so many words, repeatedly, in Section 2.
 - As Earth is a noun, please capitalize the first letter of the word wherever it appears.

**Summary Of The Paper:**

The paper presents a new PyTorch library, TorchGeo, specifically for processing deep learning models on geospatial datasets. The library majorly provides a variety of benchmark geospatial datasets, along with pre-trained models on multi-spectral and SAR images. The library also provides a set of pre-processing steps which can pixel-align patches of data on-the-fly. Apart from these major contributions, the library has functions for augmenting (transforming) and sampling geospatial datasets. The paper is overall, well written and easy to understand.

**Summary Of The Review:**

The paper makes processing geospatial datasets easier with deep learning algorithms, and is worthy of publication at another venue after responding to some concerns. The paper does not contribute to any aspect of representation/deep learning, and hence ICLR would not be a right fit. The contribution, the TorchGeo package, is more relevant for the remote sensing field.

---

> ### Author Response · Authors · 2021-11-16
> **Rebuttal**
>
> Thank you for the thorough review, we believe your comments have helped significantly improve the paper.
>
> > The authors state that the library is also compatible with PyTorch dataloaders, which is good
>
> We agree! We’d like to emphasize that TorchGeo is an open source library and is (by design) similar to the popular “torchvision” library. Users of torchvision should be able to use TorchGeo without having to learn many new concepts.
>
> > My major concern is what authors mention in the second last line of the Introduction, (also evident from Tables 3 & 4) - “ImageNet pretraining significantly improves spatial generalization performance” ...
>
> Our hypothesis is that ImageNet pre-trained weights are more diverse than the weights/filters that a network learns having been trained solely on data from one domain (e.g. on imagery from Delaware alone) which allows them to use _some_ relevant features from multiple domains. We do not have any evidence to support this, so did not speculate further in the text, however the effect is not an effect of hyperparameter tuning as both the random init and ImageNet pre-trained model were tuned for their validation performance on the Delaware set—neither model had knowledge of any out-of-domain data/performance during training.
>
> Determining why training from ImageNet weights leads to more generalizable models, especially when using remotely sensed imagery, is a very interesting research project, but we believe it is outside of the scope of this paper. We have added relevant citations [1, 2] to this section to give context to this line of experimentation. We also note that this line of research is exactly the type that TorchGeo is made to help explore.
>
> [1] Hendrycks, Dan, Kimin Lee, and Mantas Mazeika. "Using pre-training can improve model robustness and uncertainty." In International Conference on Machine Learning, pp. 2712-2721. PMLR, 2019. http://proceedings.mlr.press/v97/hendrycks19a/hendrycks19a.pdf
>
> [2] Huh, Minyoung, Pulkit Agrawal, and Alexei A. Efros. "What makes ImageNet good for transfer learning?." arXiv preprint arXiv:1608.08614 (2016). https://arxiv.org/pdf/1608.08614.pdf
>
> >  I believe providing benchmark geospatial datasets would be incomplete without providing access to some ISPRS benchmarks (https://www.isprs.org/education/benchmarks.aspx) such as Potsdam and Vaihingen datasets, especially in Section 4.1 and Table 3.
>
> Thank you for the suggestion. We agree and have implemented the Potsdam and Vaihingen datasets (specifically, the 2D semantic segmentation tasks from WG II/4) to start with. We hadn’t previously implemented these as the datasets are not able to be downloaded automatically (users need to fill out a form and download the datasets over FTP), while most of our current _benchmark_ datasets have an auto-download functionality (similar to the torchvision library). We’ve also implemented several other datasets, mentioned in the general response.

---

> ### Author Response · Authors · 2021-11-16
> **Rebuttal 2**
>
> > Table 2: Please give references to the labelled datasets (CDL, Chesapeak Land Cover etc.)
>
> Thank you for catching this. We've added citations for all datasets in Table 2.
>
> > Table 3: The table does not provide pre-trained models on multispectral or SAR images. It only provides pre-trained models for mostly RGB imageries. As your contribution is towards geospatial datasets such as multispectral and SAR images, you should also provide pre-trained models on these datasets. Example: datasets presented in Table 1.
>
> We've added experimental results for 3 new datasets and clarified in Table 3 that 4 out of 7 of these datasets include MSI and SAR data. See the general response for more details.
>
> > Section 3.2 and 4.2: Does any past work, such as in GDAL, also implement sampling methods? Can you compare the performance of your methods, their utility, as compared to the past methods?
>
> As far as we know, no existing libraries aim to solve the problem of sampling pixel-aligned chips/patches of data from larger rasters.
>
> * GDAL (Geospatial Data Abstraction Library) is a low-level library that provides methods for loading and transforming vector and raster data stored in a variety of different data formats as well as performing operations over that data. Some file formats that GDAL can interpret, like COG and (recently) Zarr, allow for the efficient sampling of data in time proportional to the area to be sampled (rather than in time proportional to the total raster size). Our library would not be possible without these, however there is nothing to directly compare to.
> * RasterVision assumes that the (raster) data layers that you would like to use are already preprocessed to be pixel-aligned, and then performs the sampling step in terms of pixel coordinates. This is a best case scenario and will be similar to the “Preprocessed, Cached” experiment in Section 4.2.
>
> > The paper does not contribute to any aspect of representation/deep learning, and hence ICLR would not be a right fit. The contribution, the TorchGeo package, is more relevant for the remote sensing field.
>
> See our general response to this question above. We further think this paper is specifically suitable for the computer science community (versus the remote sensing community) as it makes it easier for researchers to use remote sensing data (a topic already familiar to the remote sensing community) with tools that they may already be familiar with.
>
> > Minor comments: Please cite Figure 1 and Table 2 in the main text; There’s no need to embolden so many words, repeatedly, in Section 2; As Earth is a noun, please capitalize the first letter of the word wherever it appears
>
> We have addressed these.

---

> ### Comment · Reviewer_stxS · 2021-11-30
> **Response to Rebuttals**
>
> Thank you for the detailed answers and for providing additional datasets.
>
> TorchGeo would certainly be a good software platform for remote sensing applications. However, I am not sure if it addresses representation learning. Hence, I leave it to the ACs' discretion to decide whether it would be a good fit to ICLR or not.
>
> Other than that, I am fine with the rebuttals and the paper in general.

---

### Official Review · Reviewer_xaxq · 2021-11-03

**Correctness:** 4
**Technical Novelty And Significance:** 2
**Empirical Novelty And Significance:** Not applicable
**Recommendation:** 5
**Confidence:** 5

**Main Review:**

Paper review for manuscript “TORCHGEO: DEEP LEARNING WITH GEOSPATIAL DATA”

This paper describes a newly created library for a geospatial data processing. The library is based on PyTorch and combines the functionality of common geospatial libraries.


The proposed Python package is designed to facilitate usage of deep learning modles with geospatial data. TorchGeo provides data loaders for common geospatial datasets, composable data loaders for arbitrary geospatial raster and vector data, samplers appropriate for geospatial data, models, transforms, and model trainers. TorchGeo allows users to bypass the common pre-processing steps necessary to align geospatial data with imagery and performs this processing on-the-fly. We benchmark TorchGeo dataloader speed and demonstrate how TorchGeo can be used to create reproducible benchmark results in several geospatial datasets. TorchGeo can serve as a platform for performing geospatial machine learning research.

Main paper contributions are stated to be as following:
1. providing data loaders for common geospatial datasets from the literature;
2. composable data loaders for arbitrary geospatial raster and vector data with the ability to create pixel-aligned patches of data on-the-fly;
3. augmentations that are appropriate for multispectral imagery;
4. data samplers appropriate for geospatial data.

== Comparison to previous work ==
Previously, the same result (loading the data, aligning the bands, loading datasets) could be achieved by using a combination of a few packages, including gdal, rasterio and geopandas and common Python libraries (TF/Pytorch, etc.). The authors propose a more convenient way to do the same things, combined in one useful library. This library optimises data loading and aligning.
==Theory==
Theoretical part is not included in this research, it is more engineering paper.
==Method==
The propose package includes common ML methods, used with GIS data: U-net, Res net etc. Some models are pre-trained on Image net, which improves their performance.
=== Pros ===
This is a great contribution, useful for beginner researchers in the field of ML for Earth Observation data. It provided useful tools for working with EO data and loading the dataset with an improved speed.
===Cons===
The novelty of the proposed research is limited since it presents a package for data loading and model training rather than a new research method, a compilation of approaches presented before.

=== Basis for recommendation ===
I am borderline regarding this paper since it has limited scientific novelty. This is a good software engineering paper, however I’m not sure if this kind of paper is suitable for the main ICLR conference.
=== Questions to address in the revision ===
I suggest authors remove the “Transforms” part from section 3 (page 5, top) since it presents Index calculation (which is are simple arithmetic operations on numpy arrays of different combination of bands): (a) it’s too simplistic for a serious research paper; (b) Please decide if you want to use word “index” in explaining NDVI and other indices or not, currently it’s inconsistent.
=== Minor comments and additional feedback (not necessary to address in the revision)===
p.1 “composable data loaders” requires explanation (or reference to the section when t’s explained)





**Summary Of The Paper:**

This paper describes a newly created library for a geospatial data processing. The library is based on PyTorch and combines the functionality of common geospatial libraries.

**Summary Of The Review:**

I believe that this is a great work and useful tool for scientific research, but I doubt that it is suitable for ICLR main conference: it's rather a software engineering paper than a novel research.

---

> ### Author Response · Authors · 2021-11-16
> **Rebuttal**
>
> Thank you for your careful review.
>
> > This is a good software engineering paper, however I’m not sure if this kind of paper is suitable for the main ICLR conference.
>
> Thank you! Since we saw this reaction from multiple reviewers, we've added a general response above. Let us know if you have any additional concerns beyond what we've addressed.
>
> > I suggest authors remove the “Transforms” part from section 3
>
> We edited the "Transforms" section so that the word "index" is used solely to describe indexing into a geospatial dataset. We don't want to remove this section entirely since transforms are a core component of the library implementation.
>
> >  “composable data loaders” requires explanation
>
> We've clarified the meaning of "composable data loaders" in the enumerated list at the end of Section 1.
>
> Finally, we’ve made significant other improvements throughout. Please see our responses to the other reviewers.

---

### Author Response · Authors · 2021-11-16
**General Rebuttal**

We've received a few comments that were common across multiple reviewers, so we've summarized those below instead of repeating them in individual responses.

> ICLR may not be a good fit for this paper

According to the [ICLR homepage](https://iclr.cc/), the conference explores many relevant topics, including "representation learning for computer vision", "implementation issues", "software platforms", and domain-specific "applications". Since this paper touches on all of these topics, we strongly believe that this paper is suitable for ICLR. The main contribution of this paper is to introduce a software platform for representation learning on geospatial data. At the moment, it is challenging to make research progress at the intersection of remote sensing and deep learning without being an expert in both fields. TorchGeo allows remote sensing experts without deep learning experience, and deep learning researchers without remote sensing expertise, to tackle very important problems in applications like precision agriculture, natural disaster monitoring, and climate change. Although novel research ideas are important for breakthroughs in science, reproducibility and software platforms are just as important for enabling progress in these fields. In addition to this we introduce novel methods for composing and sampling from geospatial data sources in disparate coordinate systems and spatial resolutions, and reproducible results on a variety of geospatial datasets for future work to build off of.

> Would like to see additional datasets. Would like to see more experimental results on MSI and SAR data, not just RGB imagery.

Thanks for the dataset suggestions. Since our initial submission, we (and various open-source contributors) have implemented 8 new datasets:

* BigEarthNet (Sentinel 1 + 2)
* Imagery from the Seasonal Contrast paper (Sentinel 2)
* SpaceNet 4 (WorldView 2)
* SpaceNet 7 (Planet Lab Dove)
* xView2 (Maxar)
* ISPRS Potsdam (aerial imagery)
* ISPRS Vaihingen (aerial imagery)
* OSCd (Sentinel-2)

These datasets have all been added to Table 1. We've also added citations to Table 2. In addition, we've added experimental results for 3 new datasets:

* ETCI 2021 (Sentinel-1, SAR)
* EuroSat (Sentinel-2, MSI)
* UC Merced (RGB)

We've also clarified that the Chesapeake CVPR (NAIP) and So2Sat (Sentinel-2) datasets are both MSI. The updated Table 3 shown in the paper.

> If I was loading and merging L8 and an S2 dataset, what would happen?

This is a great point, and something we failed to highlight in the paper. There are many different ways in which two or more generic datasets could be merged. For example, users may want to:

1. Combine image and target labels and sample from both simultaneously (e.g. Landsat 8 + Cropland Data Layer labels)
2. Combine imagery from multiple sensors (e.g. Landsat 7 + 8, or Landsat 8 + Sentinel 2)
3. Combine datasets from disparate geospatial locations (e.g. NY + PA)

TorchGeo supports all three of these use cases, and we've updated the paper to reflect this. See Appendix 3 for a more advanced example where we take the union of Landsat 7 and 8 and then take the intersection with CDL. For your specific question, in case 2 we would return image patches where all Landsat 8 and Sentinel 2 channels are stacked together (and resampled to the highest spatial resolution). This use case is especially useful for applications like multimodal learning [1] and data fusion [2].

[1] Baltrušaitis, Tadas, Chaitanya Ahuja, and Louis-Philippe Morency. "Multimodal machine learning: A survey and taxonomy." IEEE transactions on pattern analysis and machine intelligence 41, no. 2 (2018): 423-443. https://doi.org/10.1109/TPAMI.2018.2798607

[2] Schmitt, Michael, and Xiao Xiang Zhu. "Data fusion and remote sensing: An ever-growing relationship." IEEE Geoscience and Remote Sensing Magazine 4, no. 4 (2016): 6-23. https://doi.org/10.1109/MGRS.2016.2561021

---

### Decision · Program_Chairs · 2022-01-20

**Decision:**

Reject

**Comment:**

This paper develops a Python library for geospatial data based on Pytorch, TorchGeo. TorchGeo is a useful tool for applying deep learning methods to geospatial data. The reviewers agrees the contribution of this library. It will help machine learning researchers to use geospatial data and help geospatial researchers to apply machine learnig methods. However, the technical contribution is low, and the novelty is not high enough since the results can be achieved by a combination of existing packages.